# The complex Liouville string: The gravitational path integral

Scott Collier[1], Lorenz Eberhardt[2] and Beatrix Mühlmann[3]

**1** Center for Theoretical Physics, Massachusetts Institute of Technology,
Cambridge, MA 02139, USA
**2** Institute for Theoretical Physics, University of Amsterdam,
PO Box 94485, 1090 GL Amsterdam, The Netherlands
**3** School of Natural Sciences, Institute for Advanced Study, Princeton, NJ 08540, USA

⋆ sac@mit.edu , † l.eberhardt@uva.nl , ‡ beatrix@ias.edu

## Abstract

We give a rigorous definition of sine dilaton gravity in terms of the worldsheet theory of the complex Liouville string [1]. The latter has a known exact solution that we leverage to explore the gravitational path integral of sine dilaton gravity – a quantum deformation of dS JT gravity that admits both $AdS_2$ and $dS_2$ vacua. We uncover that the gravitational path integral receives contributions from new saddles describing transitions between vacua in a third-quantized picture. We also discuss the sphere and disk partition function in this context and contrast our findings with other recent work on this theory.



# 1 Introduction

Unlike the situation with negative cosmological constant, there is a scarcity of theoretically tractable models of quantum gravity with a positive cosmological constant. Given that our own universe is dominated by a tiny yet positive cosmological constant and the challenges faced by top-down constructions of de Sitter quantum gravity in string theory, finding and understanding such models is an extremely pressing theoretical issue.

In the literature there is an ongoing attempt to exploit the calculable control of two-dimensional theories to model de Sitter quantum gravity [2–12]. Despite the progress that has been made, a unifying picture has remained elusive. The case of two-dimensional gravity is particularly tractable due to the persistent paradigm of precise dualities involving double-scaled matrix models as dual descriptions. These dualities often arise from viewing the bulk gravity theory as the worldsheet of a string theory, which allows one the use of the well-developed technology of string perturbation theory. These models are technically much more tractable than their higher-dimensional counterparts, yet retain some of the essential physical characteristics present in higher spacetime dimensions.

Recently AdS$_2$ JT gravity has been embedded into this paradigm [13]; both the $p \to \infty$ limit of the $(2, p)$ minimal string [14] and the $b \to 0$ limit of the Virasoro minimal string [15] constitute stringy realizations of AdS$_2$ JT gravity. The dual description is a double-scaled Hermitian matrix integral.

These type of gravity/matrix integral dualities often involve theories with a negative cosmological constant in the bulk, making them low-dimensional exemplars of the AdS/CFT paradigm. Adopting the calculable control in low spacetime dimensions to construct a model for dS quantum gravity and in particular hints of a microscopic completion has been investigated in recent works. Guided by the success of AdS$_2$ JT gravity, and the Nariai limit of the 4d Schwarzschild dS black hole, the analytic continuation to dS$_2$ JT gravity has been studied in [2–4, 12, 16–18] among others. Deformations of the dilaton potential in such a way that one obtains interpolating geometries (such as the so-called centaur geometries) between AdS$_2$ and dS$_2$, with the hope to borrow insights from the microscopic picture of AdS$_2$ for de Sitter, have also been investigated in [17, 19]. A specific double scaling limit of the SYK model has also recently been interpreted as a theory of dS gravity [10, 11, 20–24].

Constructing concrete 2d string theories with (worldsheet) de Sitter vacua is thus an interesting challenge. In this work we propose an explicit model depending on a parameter $b^2 \in i\mathbb{R}$ that is realized on the worldsheet in terms of two coupled Liouville theories of central charge $c = 13 \pm 6(b^2 + b^{-2})$. We studied this so-called complex Liouville string in our previous papers [25–27]. As we already discussed in [26], the theory reduces to de Sitter JT gravity on the worldsheet in the semiclassical limit $b^2 \to i\infty$. Contrary to previous discussions of dS JT gravity where the definition of the gravitational path integral requires a substantial amount of guesswork, the stringy realization of the theory gives a completely rigorous way to define the path integral and thus guides our understanding of it. It is thus an ideal playground in which to explore 2d de Sitter gravity. Away from the semiclassical $b^2 \to i\infty$ limit, it will turn out that the theory is actually much richer than dS JT gravity since it admits vacua of both positive and negative cosmological constant.

This paper is part of a series of papers [25–27] and an expanded version of the corresponding section of [1]. The main theme of this paper is to apply the technical control over the theory developed in our previous papers to extract lessons about the gravitational path integral of 2d dilaton gravity. We first show that the complex Liouville string [25] admits a path integral description in terms of a 2d dilaton gravity theory with a sine potential. This theory has come to be known as sine dilaton gravity, and has been studied recently from a variety of points of view [22, 23, 28]. When studied on hyperbolic surfaces this theory admits classi-

cal solutions with both signs of the cosmological constant, loosely reminiscent of the centaur geometries of [2, 17]. We reproduce the qualitative structure of the string amplitudes of the complex Liouville string that we uncovered in [25, 26] via the two-dimensional gravitational path integral and understand some of its features from a gravitational perspective. While one can see the correct structure emerging, we have to use the worldsheet result in several places to inform us how to proceed. In particular, the worldsheet answer dictates that there are a multitude of contributing saddles of the gravitational path integral. Perhaps most surprisingly, we uncover new saddles for which the worldsheet (i.e. spacetime) degenerates to a nodal surface with different vacua on the different components. Such solutions can be interpreted as transition amplitudes between vacua corresponding to universes with possibly different cosmological constants. The worldsheet solution moreover implies a specific recombination of certain saddles with reflected values of the dilaton that is difficult to explain directly from the two-dimensional gravitational path integral. Since the complex Liouville string is dual to a two-matrix integral [26] this discussion in particular also provides an explicit and precise realization of a duality between a 2d gravity theory that admits de Sitter vacua and a matrix integral.

We also discuss various topologies of lower complexity such as the two-sphere and torus partition function, as well as the Euclidean black hole (the disk partition function) whose gravitational path integral is qualitatively different. In particular we analyze the divergence of the sphere partition function and comment on the thermodynamics of the Euclidean black hole. We then interpret our results in terms of a third-quantized or universe field theory where the uncovered saddles lead to transitions between dS and AdS vacua.

**Outline.** We start in section 2 by explaining the relation between the complex Liouville string and sine dilaton gravity. We then reproduce and reinterpret the explicit formulas for the string amplitudes $\mathsf{A}_{g,n}^{(b)}$ from the gravitational path integral. In section 3 we discuss in detail the gravitational path integral on a few interesting exceptional surfaces that do not fit into the general treatment of the previous discussion. We explore the consequences for 2d de Sitter quantum gravity in section 4.

## 2 Sine dilaton gravity

We consider a worldsheet theory of two coupled complex Liouville theories of central charges $c = 13 \pm i\lambda$:

$$\begin{array}{ccc} \text{Liouville CFT} & \text{(Liouville CFT)}^* & \mathfrak{b}\mathfrak{c}\text{-ghosts} \\ c = 13 + i\lambda & \oplus & c^* = 13 - i\lambda & \oplus & c_{\text{gh}} = -26 \end{array}, \quad (2.1)$$

which is known as the complex Liouville string ($\mathbb{C}$LS). We will now explain some features of this coupled Liouville theory from the path integral perspective and explain its relation to a two-dimensional dilaton gravity theory with a sine potential.

### 2.1 Mapping of the worldsheet action

We start by rewriting the action of the worldsheet theory.

**Complex Liouville string.** The action of the $c = 13 + i\lambda$ Liouville theory is given by

$$S[\varphi] = \frac{1}{4\pi} \int_{\Sigma_{g,n}} \mathrm{d}^2 x \sqrt{\tilde{g}} \left( \tilde{g}^{\mu\nu} \partial_\mu \varphi \partial_\nu \varphi + Q \widetilde{\mathcal{R}} \varphi + 4\pi \mu \, \mathrm{e}^{2b\varphi} \right), \quad (2.2)$$

where the Liouville central charge is related to the parameters $Q$ and $b$ as following

$$c = 1 + 6Q^2, \qquad Q = b^{-1} + b, \qquad b \in e^{\frac{\pi i}{4}} \mathbb{R}_+. \tag{2.3}$$

The second Liouville CFT is the complex conjugate of (2.2) with the Liouville fields $\varphi$ and $\varphi^*$ complex conjugates of each other

$$S^*[\varphi] = \frac{1}{4\pi} \int_{\Sigma_{g,n}} d^2x \sqrt{\tilde{g}} \left( \tilde{g}^{\mu\nu} \partial_\mu \varphi^* \partial_\nu \varphi^* + Q^* \widetilde{\mathcal{R}} \varphi^* + 4\pi\mu^* e^{2b^* \varphi^*} \right). \tag{2.4}$$

We indicate the background metric and the Ricci scalar with a tilde; $\mu$ is the cosmological constant in the sense of Liouville theory. Thus the total worldsheet action without ghosts is simply given by $S[\varphi] + S^*[\varphi] = 2 \operatorname{Re} S[\varphi]$. This leads to a critical string theory. In particular the choice of background metric is immaterial.

**Sine dilaton action.** Following the logic of [15,29], we now perform the change of variables

$$b\varphi = \rho + i\pi\Phi, \qquad b^*\varphi^* = \rho - i\pi\Phi, \tag{2.5}$$

where $\Phi$ and $\rho$ are real scalar fields. We will interpret $\rho$ as the Weyl factor $g = e^{2\rho} \tilde{g}$ of a physical metric. We then rewrite the total worldsheet action $S[\varphi] + S^*[\varphi]$ as

$$S[\Phi, \rho] = \frac{i}{2b^2} \int_{\Sigma_{g,n}} d^2x \sqrt{\tilde{g}} \left( \Phi(\widetilde{\mathcal{R}} - 2\widetilde{\nabla}^2 \rho) + \frac{1}{\pi} e^{2\rho} \sin(2\pi\Phi) \right)$$
$$+ \frac{i}{b^2} \int_{\partial\Sigma_{g,n}} dx \sqrt{\tilde{h}} \, \Phi \partial_n \rho + \frac{1}{4\pi} \int_{\Sigma_{g,n}} d^2x \sqrt{\tilde{g}} \, (2\widetilde{\mathcal{R}} \rho). \tag{2.6}$$

The sine potential arises as follows: The combination of the Liouville potentials under the change of variables (2.5) is

$$\mu e^{2b\varphi} + \mu^* e^{2b^*\varphi^*} \to e^{2\rho} \left( i(\mu - \mu^*) \sin(2\pi(-ib^2)\Phi) + (\mu + \mu^*) \cos(2\pi(-ib^2)\Phi) \right), \tag{2.7}$$

where $(-ib^2) \in \mathbb{R}_+$. By shifting $\varphi$ appropriately in (2.2) and (2.4), we can set $\mu$ without loss of generality to any non-zero value. It is convenient for the following to choose $\mu = \frac{1}{4\pi b^2} \in i\mathbb{R}$. This gives the convenient normalization of the sine potential that we used in (2.6).

We see that the first line of (2.6) has a prefactor $\frac{i}{b^2}$ which indicates that we should think of $\hbar \sim -ib^2$. Thus we will call the limit $(-ib^2) \to 0$ the semiclassical limit. In this limit, the part of the action depending solely on the Weyl factor $\rho$ in the second line of (2.6) is subleading. It is part of the anomaly contribution (with $Q = 2$ consistent with a $c = 25$ Liouville type theory) that arises when we Weyl gauge fix the following sine dilaton action

$$S[\Phi, g] = \frac{i}{2b^2} \int_{\Sigma_{g,n}} d^2x \sqrt{g} \left( \Phi \mathcal{R} + \frac{\sin(2\pi\Phi)}{\pi} \right), \tag{2.8}$$

and use the relation between the Ricci scalar of the fiducial metric and the physical metric

$$\mathcal{R} = e^{-2\rho}(\widetilde{\mathcal{R}} - 2\widetilde{\nabla}^2 \rho). \tag{2.9}$$

Moreover, provided we conveniently choose a background metric with vanishing extrinsic curvature, the boundary term in (2.6) is proportional to the extrinsic curvature $K$. We call the theory governed by the bulk action (2.8) sine dilaton gravity.

## 2.2 Vertex operators

Let us next discuss how vertex operators behave under this mapping.

**Vertex operators of the $\mathbb{C}$LS.** The vertex operators $V_p$ of $c = 13 + i\lambda$ Liouville theory are labelled by their conformal dimension $h$, parametrized as

$$h = \tilde{h} = \frac{c-1}{24} - p^2 . \tag{2.10}$$

Reality of the stress tensor of the combined Liouville theories (2.1) implies that the conformal dimension of the $c^* = 13 - i\lambda$ Liouville CFT is the complex conjugate of (2.10), i.e. $h^- = h^*$. The minus superscript denotes variables in the Liouville CFT with central charge $c^*$. Together with the on-shell condition $h + h^- = 1$ of string theory, this implies

$$p^2 \in i\mathbb{R} , \qquad p^- = \pm i p , \tag{2.11}$$

as the physical state condition, i.e. the Liouville momenta are rotated by 45 degrees in the complex plane. We will usually assume that

$$p \in e^{-\frac{\pi i}{4}} \mathbb{R}_+ , \qquad p^- \in e^{\frac{\pi i}{4}} \mathbb{R}_+ , \tag{2.12}$$

which can also be stated as

$$bp \in \mathbb{R}_+ , \qquad b^- p^- \in \mathbb{R}_+ . \tag{2.13}$$

**Path integral.** We next interpret these vertex operators in a path integral language. For this, we have to be somewhat careful about the normalization. In [25], we adopted a normalization for which the two point functions of Liouville CFT primaries $\langle V_p(0) V_{p'}(1) \rangle$ are normalized as $\rho_b(p)^{-1} \delta(p - p')$, where $\rho_b(p) = 4\sqrt{2} \sin(2\pi b p) \sin(2\pi b^{-1} p)$. It is in particular invariant under reflections $p \to -p$ of the Liouville momenta, unlike the case for the usual exponential operators $e^{2\alpha\varphi}$ of Liouville theory. This can be achieved in the path integral formalism by considering the following reflection-symmetric combination of appropriately normalized vertex operators [30, 31],

$$V_p = S^{(b)}(p)^{-1/2} \rho_b(p)^{-\frac{1}{2}} e^{(Q-2p)\varphi} + S^{(b)}(p)^{1/2} \rho_b(p)^{-\frac{1}{2}} e^{(Q+2p)\varphi} , \tag{2.14}$$

where $S^{(b)}(p)$ is the Liouville reflection coefficient

$$S^{(b)}(p) \equiv -\left(\pi \mu \gamma(b^2)\right)^{\frac{2p}{b}} \frac{\Gamma(1 - 2b^{-1}p)\Gamma(1 - 2bp)}{\Gamma(1 + 2b^{-1}p)\Gamma(1 + 2bp)} , \tag{2.15}$$

with $\gamma(x) \equiv \Gamma(x)/\Gamma(1-x)$. This leads to a convenient convention for the Liouville CFT three point structure constant $C_b(p_1, p_2, p_3)$, which differs from the more familiar DOZZ conventions [32–34] as follows [35]

$$C_b(p_1, p_2, p_3) = \left( \frac{(\pi \mu \gamma(b^2) b^{2-2b^2})^{\frac{Q}{2b}}}{2^{\frac{3}{4}} \pi} \frac{\Gamma_b(2Q)}{\Gamma_b(Q)} \right) \frac{C_{\text{DOZZ}}(p_1, p_2, p_3)}{\sqrt{\prod_{j=1}^3 S^{(b)}(p_j)\rho_b(p_j)}} , \tag{2.16}$$

where $\gamma(x) = \frac{\Gamma(x)}{\Gamma(1-x)}$. The $p$-independent prefactors in parentheses will not be important in what follows. In the string theory, we consider the combination $\mathcal{N}_b(p) V_p^+ V_{ip}^-$ with $\mathcal{N}_b(p)$ a

convenient leg factor whose form can be found in [25]. Thus in the path integral formalism such vertex operators take form

$$
\mathcal{V}_p = \mathcal{N}_b(p) V_p^+ V_{ip}^- \supset \mathcal{N}_b(p) \frac{e^{2\left(\rho + \frac{\pi i}{b^2}\Phi\right) - 4\pi i \frac{p}{b}\Phi}}{\sqrt{|\rho_b(p)S^{(b)}(p)|^2}} + \mathcal{N}_b(p) \sqrt{\left|\frac{S^{(b)}(p)}{\rho_b(p)}\right|^2} e^{2\left(\rho + \frac{\pi i}{b^2}\Phi\right) + 4\pi i \frac{p}{b}\Phi}
$$

$$
\approx \mathcal{N}_{bp,-} e^{2\left(\rho + \frac{\pi i}{b^2}\Phi\right) - 4\pi i \frac{p}{b}\Phi} + \mathcal{N}_{bp,+} e^{2\left(\rho + \frac{\pi i}{b^2}\Phi\right) + 4\pi i \frac{p}{b}\Phi}, \tag{2.17}
$$

where $\mathcal{N}_{bp,\pm}$ are normalization factors that depend on the combination $bp$ which we keep fixed in the semiclassical limit. Additionally it was important that the cosmological constant that enters in (2.15) is purely imaginary and hence $\mu^* = -\mu$.

**A complex Seiberg bound.** Notice that we only included two out of the four possible exponentials in (2.17). The combination $\mathcal{N}_b(p) V_p^+ V_{ip}^-$ would in principle also include the exponentials

$$
\mathcal{V}_p \supset \tilde{\mathcal{N}}_{bp,+} \mu^{\frac{2p}{b}} e^{2(\rho + \frac{\pi i}{b^2}\Phi) + \frac{4p}{b}\rho} + \tilde{\mathcal{N}}_{bp,-} \mu^{-\frac{2p}{b}} e^{2(\rho + \frac{\pi i}{b^2}\Phi) - \frac{4p}{b}\rho}. \tag{2.18}
$$

One important difference between the exponentials in (2.17) and (2.18) is the presence of powers of the Liouville cosmological constant $\mu$ in the latter. Naively, following the KPZ scaling [36] of the correlators built out of these vertex operators in (2.20), the dependence on the momentum $p$ in the exponent of $\mu$ should drop out. This is a similar situation as in standard Liouville theory, where vertex operators below the Seiberg bound $\alpha = \frac{Q}{2} - p < \frac{Q}{2}$ should only have one of the two exponentials since they don't admit a plane wave interpretation [37]. This is indicated by the fact that the two terms have two different real $\mu$ exponents and would violate KPZ scaling [36]. The present situation is similar as two out of the four terms would lead to a different KPZ scaling, meaning that we should only consider the two terms in (2.17). As a further piece of evidence for this interpretation, the $\rho$ dependence in (2.18) would lead to a non-diffeomorphism invariant term in the metric formalism. Indeed, the term $e^{2\rho}$ in (2.17) precisely combines with the measure factor $\sqrt{\tilde{g}}$ to $\sqrt{g}$ and is thus rewritten as

$$
\int d^2 x \sqrt{\tilde{g}} \mathcal{V}_p = \int d^2 x \sqrt{g} \left( \mathcal{N}_{bp,-} e^{\frac{2\pi i}{b^2}\Phi - 4\pi i \frac{p}{b}\Phi} + \mathcal{N}_{bp,+} e^{\frac{2\pi i}{b^2}\Phi + 4\pi i \frac{p}{b}\Phi} \right), \tag{2.19}
$$

which is in particular diffeomorphism-invariant.

## 2.3 Gravitational path integral

We want to compare the gravitational path integral of the sine dilaton gravity theory with the worldsheet [25] and matrix integral [26] string amplitudes. For this we study the path integral of sine dilaton gravity with $n$ vertex operator insertions:

$$
Z_n^{(b)}(S_0; \boldsymbol{p}) \equiv \sum_{g \geqslant 0} e^{S_0 \chi_{g,n}} \int \frac{[\mathcal{D}g][\mathcal{D}_g \Phi]}{\text{vol}_{\text{diff}}} e^{-\frac{i}{2b^2} \int d^2 x \sqrt{g} (\Phi \mathcal{R} + \frac{\sin(2\pi\Phi)}{\pi})} \mathcal{V}_{p_1} \cdots \mathcal{V}_{p_n}
$$

$$
= \sum_{g \geqslant 0} e^{S_0 \chi_{g,n}} Z_{g,n}^{(b)}(\boldsymbol{p}). \tag{2.20}
$$

Here $\chi_{g,n} = 2 - 2g - n$ is the Euler characteristic associated with a Riemann surface of genus $g$ with $n$ punctures. We consider compact surfaces for now and hence discard the boundary term in (2.8). We denoted these quantities in the string theory language in [25,26] by $A_n^{(b)}(S_0; \boldsymbol{p})$, since they corresponded to string theory amplitudes (summed over genera) in that language. Since we now think of them as a gravitational path integral, and to distinguish them from the string theory result, we will denote them in the following by $Z_n^{(b)}(S_0; \boldsymbol{p})$. In this

correspondence, the Weyl factor of the metric $g$ is mapped to a combination of the Liouville fields and consequently only the diffeomorphism symmetry is gauged in the gravitational language (2.20). We will first discuss the case where $\chi_{g,n} < 0$ and postpone the discussion of the exceptional cases $\chi_{g,n} \geqslant 0$ to section 3.

**Equations of motion.** The equations of motion of the sine dilaton theory are given by

$$\mathcal{R} + 2\cos(2\pi\Phi) = 4\pi\sum_{j=1}^{n}(1-2bp_j)\delta^2(\xi-\xi_j), \tag{2.21a}$$

$$\nabla_\mu\nabla_\nu\Phi - g_{\mu\nu}\nabla^2\Phi - \frac{1}{2\pi}g_{\mu\nu}\sin(2\pi\Phi) = 0, \tag{2.21b}$$

where for convenience we focus on one of the exponentials in the vertex operator (2.17). Using reflection symmetry we can easily incorporate the second exponential corresponding to $p_j \to -p_j$ in the source term. Notice also that the normalizations of the vertex operators in (2.17) are of order 1 in the semiclassical limit $b \to 0$ and do not influence the equations of motion.

The source terms in (2.21b) lead to curvature singularities in the metric. For $bp_j > 0$ $(< 0)$, they lead to conical defects (excesses) with defect (excess) angle $4\pi|bp_j|$. For $bp_j \in i\mathbb{R}$, they instead lead to a geodesic boundary with boundary length $\ell_j = 4\pi|bp_j|$. This is in line with the expectation coming from the quantization of Teichmüller space for real $b$ [38].

**Constant AdS and dS saddles.** The equations of motion imply that $\kappa^\mu = \epsilon^{\mu\nu}\partial_\nu\Phi$ is a Killing vector, i.e.

$$\nabla_\mu\kappa_\nu + \nabla_\nu\kappa_\mu = 0. \tag{2.22}$$

A generic surface $\Sigma_{g,n}$ does not admit any non-zero Killing vectors. Thus $\kappa^\mu = 0$ and the dilaton needs to be constant in order to solve the equations of motion. For a constant dilaton to solve the equations of motion it must correspond to a zero of the potential and thus we have

$$\Phi_* = \frac{m}{2}, \quad \mathcal{R}_* = 2(-1)^{m+1} + 4\pi\sum_{j=1}^{n}(1-2bp_j)\delta^2(\xi-\xi_j), \qquad m \in \mathbb{Z}. \tag{2.23}$$

This is quite remarkable! We have found an infinite number of classical saddles with constant dilaton labelled by an integer $m \in \mathbb{Z}$. For even $m$, the Ricci scalar $\mathcal{R}$ is negative, whereas for odd $m$ we obtain a positive scalar curvature. The sine dilaton gravity theory (2.20) thus admits a family of alternating anti-de Sitter and de Sitter vacua. Capturing vacua with different signs of the cosmological constant is vaguely reminiscent of what happens in [2,17], which consider deformations of the JT dS potential.

**Piecewise constant saddles.** Anticipating the answer for the string amplitudes $\mathsf{A}_{g,n}^{(b)}$ given in [26], we know that this is not a complete list of saddles. The worldsheet and dual matrix model inform us that there are also saddles corresponding to degenerated (nodal) surfaces, which consist topologically of two simpler Riemann surfaces connected at a single nodal point, or a single Riemann surface connected to itself at two nodal points. For such surfaces, the dilaton only needs to be piecewise constant and the value of $m$ can be different on the distinct components. Some examples of such surfaces are shown in figure 2. While the moduli of the surface were generic for the simple saddles in the previous discussion, in the degenerated solutions the moduli are restricted to the pinching limit at the boundaries of moduli space and thus this type of saddles has fewer flat directions than the generic saddle.

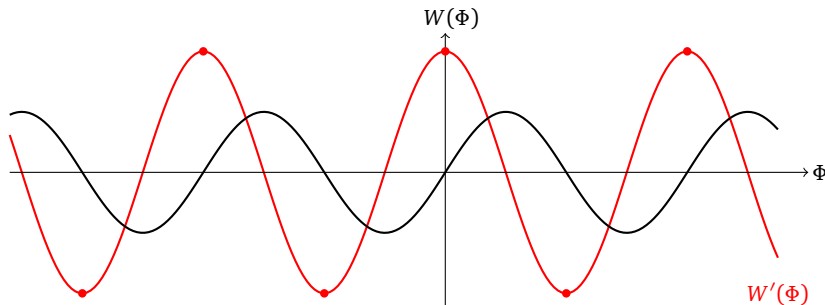

Figure 1: Indicated is the potential $W(\Phi) = \sin(2\pi\Phi)/\pi$ and its derivative. The latter sets the on-shell value of the curvature of the solutions to the equations of motion. The red dots indicate the alternating value of the scalar curvature $\mathcal{R}_* = \pm 2$ as imposed by the equations of motion (2.23).

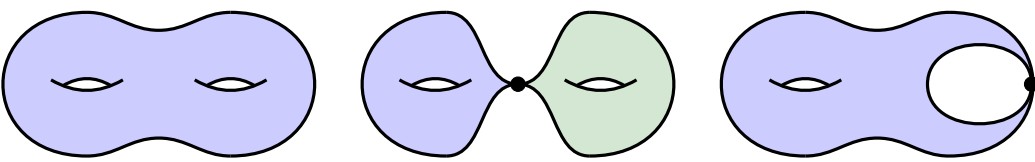

Figure 2: A selection of (possibly degenerate) saddles contributing to the genus-two path integral. The different colors indicate the possibility of different constant values for the dilaton.

In general, we can combinatorially encode all such degenerations with the help of Feynman-like graphs known as stable graphs. Every internal edge $e \in \mathcal{E}$ signifies a node, while every internal vertex represents a component of the nodal surface. Every vertex $v \in \mathcal{V}$ carries a genus $g_v$ and the value of the dilaton $m_v \in \mathbb{Z}$ as additional data. Stability of the nodal surface means that none of the connected components of the surface have a non-trivial automorphism group which would invalidate the discussion above. As in figure 2, the additional label $m_v$ can be thought of as a color and thus we call such a graph $\Gamma$ encoding a particular vacuum a colored stable graph.

This discussion in particular means from the gravitational point of view that there exist gravitational saddles which describe transitions from a dS to an AdS universe, and vice versa. We will have more to say about this in section 4.

**On-shell action.** We want to evaluate the gravitational path integral around the classical saddles that we identified above. The first step is the computation of the on-shell action. We start with the simpler case where $m$ takes a single value on the entire surface, corresponding to the trivial stable graph. To study the fluctuation theory of the sine dilaton gravity theory it is useful to go back to the formulation in Weyl gauge (2.6). The equation of motion (2.21a) in Weyl gauge and taking without loss of generality $\widetilde{R} = -2$ imply that

$$\rho_* = -\frac{1}{2} \log\left((-1)^m\right). \tag{2.24}$$

We can hence write the saddle point value as

$$\Phi_* = \frac{m}{2}, \quad \rho_* = \frac{\pi i m}{2} + \pi i s, \qquad m, s \in \mathbb{Z}. \tag{2.25}$$

Here $s$ parameterizes the ambiguity of the logarithm. In both the AdS and the dS case we consider a general complex saddle and only the AdS case $m \in 2\mathbb{Z}$ admits a real saddle. The

necessity of including complex saddles in similar computations was demonstrated for example in [39]. Importantly, the dS saddle $m \in 2\mathbb{Z}+1$ leads to a physical metric $g_{\mu\nu} = e^{2\rho_*}\tilde{g}_{\mu\nu} = -\tilde{g}_{\mu\nu}$ with $(-,-)$ signature, similarly to the dS saddles discussed in [3,12]. The on-shell action evaluates to

$$S_{\text{o.s.}}[\Phi_*,\rho_*] = \frac{\pi i m}{b^2}\chi_g + \pi i(m+2s)\chi_g, \tag{2.26}$$

where $\chi_g \equiv 2-2g$. The second term comes from the anomaly action in (2.6). In summary we obtain to leading order

$$\int \frac{[\mathcal{D}g][\mathcal{D}_g\Phi]}{\text{vol}_{\text{diff}}} e^{-\frac{i}{2b^2}\int d^2x\sqrt{g}(\Phi\mathcal{R}+\frac{\sin(2\pi\Phi)}{\pi})}\mathcal{V}_{p_1}\cdots\mathcal{V}_{p_n}$$
$$\approx (-1)^{m\chi_{g,n}}f_m e^{-\frac{\pi i}{b^2}m\chi_{g,n}}\left(\mathcal{N}_{bp,+}e^{2\pi im\sum_j \frac{p_j}{b}} + \mathcal{N}_{bp,-}e^{-2\pi im\sum_j \frac{p_j}{b}}\right), \tag{2.27}$$

where $f_m$ gives the multiplicity of the saddles labelled by $(m,s)$ (going forward we drop the label $s$ since it doesn't influence the saddle).

**Fluctuations.**     We can also study the fluctuations on top of the saddles

$$\Phi = \Phi_* + \delta\Phi, \qquad \rho = \rho_* + \delta\rho. \tag{2.28}$$

The fluctuation theory of (2.6) does not distinguish the parity of $m$ and is given for both the de Sitter and the anti-de Sitter saddles by

$$\int \frac{[\mathcal{D}\delta\Phi][\mathcal{D}\delta\rho]}{\text{vol}_{\text{diff}}} e^{-\frac{i}{b^2}\int d^2x\sqrt{\tilde{g}}\left(\delta\rho(-\tilde{\nabla}^2+2)\delta\Phi+\delta\rho^2\delta\Phi-\frac{4}{3}\pi^2\delta\Phi^3+\dots\right)}\mathcal{V}_{p_1}\cdots\mathcal{V}_{p_n} = \mathsf{V}_{g,n}^{(b)}(i\mathbf{p}). \tag{2.29}$$

The fluctuations are identical to the (analytic continuation of the) fluctuations of sinh dilaton gravity theory studied for example in [15], since the two theories only differ by a rotation of the integration contour. The path integral (2.29) thus evaluates to the string amplitudes of sinh dilaton gravity. These constitute a quantum deformation of the Weil-Petersson volumes referred to as quantum volumes, and by a similar relation to a specific worldsheet string theory as discussed here (the Virasoro minimal string), they were completely determined in [15]. Of course the fluctuations linear in $\delta\Phi$ are the fluctuations of AdS$_2$ JT gravity [13], which is consistent with the quantum volumes being the quantum extensions of the Weil-Petersson volumes. With this in mind we can evaluate the path integral (2.20) to all-loop orders. For each component surface labelled by the dilaton saddle $m$ with $n$ vertex operarator insertions we thus obtain

$$Z_{g,n}(m;\boldsymbol{p}) \equiv \int \frac{[\mathcal{D}g][\mathcal{D}_g\Phi]}{\text{vol}_{\text{diff}}} e^{-\frac{i}{2b^2}\int d^2x\sqrt{g}(\Phi\mathcal{R}+\frac{\sin(2\pi\Phi)}{\pi})}\mathcal{V}_{p_1}\cdots\mathcal{V}_{p_n} \tag{2.30}$$
$$= (-1)^{m\chi_{g,n}}f_m e^{-\frac{\pi i}{b^2}m\chi_{g,n}}\prod_{j=1}^{n}\sqrt{2}\sin(2\pi m b^{-1}p_j)\mathsf{V}_{g,n}^{(b)}(i\boldsymbol{p}).$$

To obtain the sine we combine the "incoming" and "outgoing" exponentials in (2.17) and use that by construction the normalizations $\mathcal{N}_{bp,\pm}$ after including not just the leading but the all-loop contribution should satisfy $\mathcal{N}_{bp,+}/\mathcal{N}_{bp,-} = -1$. We also chose the normalization of the vertex operators such that the sine factor has an additional $\sqrt{2}$. This can obviously be achieved by choosing a matching normalization of the vertex operators.

**Path integral around the remaining saddles.** It is now simple to also work out the path integral around the nodal surfaces as in figure 2. Let us suppose for the ease of discussion that there is only one pinching as in the middle of figure 2, but the analysis generalizes to the general case. In the path integral, we have to integrate over all metric fluctuations. There is one mode that 'unpinches' the surface, while all other modes can be viewed as metric fluctuations on the nodal surface. The unpinching integral is an integral over the size of the neck separating the two components of the surface. The size of this neck parametrizes the Hilbert space of the theory [40] and thus we can trade the integral over that mode by the insertion of a complete set of states in the Hilbert space. The insertion of such a complete set of states is achieved by using orthogonality of the vertex operators,

$$\mathbb{1} = \int \mathrm{d}p \, \frac{|\mathcal{V}_p\rangle\langle\mathcal{V}_p|}{\|\mathcal{V}_p\|^2} \,. \tag{2.31}$$

The norm of the vertex operators is computed by the two-point function $A_{0,2}^{(b)}(p, p')$, whose gravitational analogue will be discussed in section 3.1. For now, we use the result obtained from string theory that $\mathbb{1} = \int (-2p \, \mathrm{d}p) |\mathcal{V}_p\rangle\langle\mathcal{V}_p|$, since the evaluation of the two-point amplitude does not need any moduli integration, but is mostly an exercise in tracking normalizations together with the correct treatment of the residual automorphism group of the two-punctured sphere [41,42]. After this realization, we see that the on-shell actions clearly add up and the fluctuation integral factorizes into the left and the right component. Thus we have e.g. for the middle picture in figure 2 for $m_1 \neq m_2$,

$$Z_{2,0}^{(b)}(m_1, m_2) = \int_0^{e^{-\frac{\pi i}{4}} \infty} (-2p \, \mathrm{d}p) \, Z_{1,1}^{(b)}(m_1, p) Z_{1,1}^{(b)}(m_2, p) \,. \tag{2.32}$$

The range of the integral parameterizes the complete set of states of the theory. For $m_1 = m_2$, the unpinching modulus is a flat direction of the saddle and this saddle becomes part of the generic saddles with one vacuum discussed above. This is reflected by the fact that the integral over $p$ in this expression diverges.

**Summing over saddles.** Finally, to evaluate the full gravitational path integral, we should sum over all saddles. As we saw, such saddles are labelled by graphs $\Gamma$ which encode the different degenerations of the surface. Every vertex of such a graph $\Gamma$ is moreover labelled by a color $m \in \mathbb{Z}$ which specifies the vacuum of the respective component of the surface. Let us denote the set of such graphs at genus $g$ with $n$ punctures by $\mathcal{G}_{g,n}^{\mathbb{Z}}$. For a graph $\Gamma \in \mathcal{G}_{g,n}^{\mathbb{Z}}$, we can compute the corresponding gravitational path integral around that saddle by

$$Z_{g,n,\Gamma}^{(b)}(\boldsymbol{p}) = \int \prod_{e \in \mathcal{E}_\Gamma} (-2p_e \, \mathrm{d}p_e) \prod_{v \in \mathcal{V}_\Gamma} Z_{g_v, n_v}^{(b)}(m_v; \boldsymbol{p}_v) \,, \tag{2.33}$$

where we take the product over the basic partition functions on all vertices and integrate over the $p$'s assigned to intermediate edges. If two adjacent vertices have the same color, we set $Z_{g,n,\Gamma}^{(b)}(\boldsymbol{p}) = 0$. As discussed above, the reason for this is that if the colors of neighboring components coincide then the corresponding integral above does not converge due to the presence of an additional flat direction that unpinches the nodal point. Instead this contribution is associated with that of the stable graph with a vertex where the two components are connected and thus are labelled by the same color.

We finally sum over all $\Gamma$'s to obtain the full partition function. In the process, we allow for arbitrary saddle multiplicities $f_\Gamma \in \mathbb{Z}_{\geqslant 0}$. We moreover need to divide by the automorphism

factor of each graph, since this is part of the two-dimensional diffeomorphism group that we gauge.[1] Thus we finally find the genus $g$ gravitational path integral,

$$Z_{g,n}^{(b)}(\boldsymbol{p}) = \sum_{\Gamma \in \mathcal{G}_{g,n}^{\mathbb{Z}}} \frac{f_{\Gamma}}{|\text{Aut}(\Gamma)|} Z_{g,n,\Gamma}^{(b)}(\boldsymbol{p}). \tag{2.34}$$

The sum over graphs $\Gamma$ implicitly includes a sum over the colors $m_v$ of each vertex.

## 2.4 Comparison to the exact worldsheet answer

**Exact answer.** We can now compare the path integral expression (2.34) with the exact answer as was obtained from the worldsheet description using the analytic bootstrap [25, 26]. We copy it here for convenience:

$$A_{g,n}^{(b)}(p_1, \dots, p_n) = \sum_{\Gamma \in \mathcal{G}_{g,n}^{\infty}} \frac{1}{|\text{Aut}(\Gamma)|} \int' \prod_{e \in \mathcal{E}_{\Gamma}} (-2p_e \, dp_e) \prod_{v \in \mathcal{V}_{\Gamma}} \left( \frac{(-1)^{m_v+1}}{\sqrt{2} b \sin(\pi m_v b^{-2})} \right)^{2g_v - 2 + n_v}$$
$$\times \prod_{j \in I_v} \sqrt{2} \sin(2\pi m_v b^{-1} p_j) V_{g_v, n_v}^{(b)}(i\boldsymbol{p}_v). \tag{2.35}$$

In order to compare the result with the gravitational path integral, it is useful to use the duality symmetry of the string amplitudes $A_{g,n}^{(b)}(\boldsymbol{p})$, which states that the result is invariant under $b \to b^{-1}$, up to a phase. We denote the exact answer by $A_{g,n}^{(b)}(\boldsymbol{p})$ to avoid confusions with the path integral result. As the reader can see, there is a significant superficial similarity between (2.35) and (2.34), when we unpack it in terms of (2.33) and (2.30). In (2.35), we are also summing over colored stable graphs $\Gamma \in \mathcal{G}_{g,n}^{\infty}$, but the colors only take values in the positive integers, $m \in \mathbb{Z}_{\geqslant 1}$. The primed integral denotes that we discard polynomially divergent contributions to the integral. Such contributions only arise when two adjacent colors in the stable graph coincide.

**Three-punctured sphere.** We now work out some examples. We start with the three-punctured sphere. The quantum volume is trivial, $V_{0,3}^{(b)} = 1$ [15] and the only stable graph is the trivial one, leading to

$$Z_{0,3}^{(b)}(p_1, p_2, p_3) = \sum_{m \in \mathbb{Z}} (-1)^m f_m e^{-\frac{\pi i}{b^2} m} \prod_{j=1}^3 \sqrt{2} \sin(2\pi m b^{-1} p_j). \tag{2.36}$$

We can compare this with the exact result

$$A_{0,3}^{(b)}(p_1, p_2, p_3) = \sum_{m \geqslant 1} \frac{2(-1)^m}{b \sin(\pi m b^{-2})} \prod_{j=1}^3 \sin(2\pi m b^{-1} p_j). \tag{2.37}$$

From a path integral perspective there is no obvious choice for the multiplicities $f_m$, $m \in \mathbb{Z}$. The choice that most closely matches (2.37) is $f_m = 1$ for $m \geqslant 1$ and $f_m = 0$ for $m \leqslant 0$. In fact, convergence of the expression (2.36) dictates that $f_m = 0$ for $m \ll 0$. However, even with this fixing of multiplicities, the two expressions don't quite match, even though they are structurally similar.

---

[1]We define the automorphism factor to be the size of the automorphism group of the graph. This is in principle only appropriate when the colors in a graph are all the same. However, since we sum over all integers for a given vertex, we overcount e.g. the graphs appearing in (2.32) by a factor of 2, which we compensate by dividing by the automorphism group.

Figure 3: Possible degenerations of the four-punctured sphere and their interpretation in terms of stable graphs. The labels of the vertices denote the genera of the components of the surface; for more details on the notation see [26].

The exact result (2.37) tells us that the on-shell action $\mathrm{e}^{-\frac{\pi i m}{b^2}}$ gets modified to $\sin(\frac{\pi m}{b^2})^{-1}$. This is a non-perturbative effect from the gravitational path integral, but it is quite surprising from that point of view. It implies that the mapping to sine dilaton gravity is not quite true at the non-perturbative level and the saddles corresponding to $m$ and $-m$ combine in a somewhat surprising way. This is a bit similar to the emergence of reflection symmetry in Liouville theory from the path integral, but we don't understand the precise mechanism that leads to this identification. It seems to be crucial to ensure that the three-point function has the correct properties; in particular the strong-weak duality symmetry $b \to b^{-1}$ of (2.36) only holds after this modification.

The normalization factors in (2.36) and (2.37) also differ. This is at first glance not a problem since we could absorb the difference into the coupling $\mathrm{e}^{S_0}$. However, upon close examination we see that the string coupling would need to be imaginary to account for this, which would correspond to a shift $S_0 \to S_0 + \frac{\pi i}{2}$ on top of the real shift in $S_0$. The imaginary string coupling was a *choice* in [25] and was motivated by the fact that for this choice, there is a non-perturbative completion of the perturbative genus expansion in form of the matrix model. Thus this disagreement is built in and should not worry us at the moment. We will discuss the implications of the imaginary string coupling from the point of view of the gravitational path integral in section 4.1.

**Four-punctured sphere.** Let us further illustrate these formulas with the four-punctured sphere. On top of the trivial stable graph, corresponding to the surface on the left of figure 3, the surface can also degenerate as in the right of the figure. The analysis of the trivial graph is basically identical to the analyis of the three-punctured sphere, so we will discuss the second contribution. Two three-punctured spheres are then glued together along an internal momentum

$$Z_{\underset{m_1\ m_2}{\circ\!-\!\circ}}^{(b)}(p_1, p_2, p_3, p_4) = \int (-2p\mathrm{d}p)\, Z_{0,3}^{(b)}(m_1; p_1, p_4, p) Z_{0,3}^{(b)}(m_2; p, p_2, p_3). \tag{2.38}$$

We suppressed the labels on the external legs of the stable graph and there are two more contributions obtained by permuting $p_1, \dots, p_4$. Each three-punctured sphere is labelled by its dilaton saddle (2.23), $m_1$ and $m_2$ respectively. As discussed above, we should have $m_1 \neq m_2$. After summing over colors and including degeneracies we thus obtain from the path integral

$$Z_{\circ\!-\!\circ}^{(b)}(\boldsymbol{p}) = \sum_{\substack{m_1, m_2 \in \mathbb{Z} \\ m_1 \neq m_2}} f_{m_1, m_2} \prod_{j=1}^{2} (-1)^{m_j} \mathrm{e}^{-\frac{i\pi}{b^2} m_j} \prod_{j=1,4} \sin(2\pi m_1 b^{-1} p_j) \prod_{j=2,3} \sin(2\pi m_2 b^{-1} p_j)$$

$$\times \int (-2p\mathrm{d}p) \sin(2\pi m_1 b^{-1} p) \sin(2\pi m_2 b^{-1} p)$$

$$= \sum_{\substack{m_1, m_2 \in \mathbb{Z} \\ m_1 \neq m_2}} f_{m_1, m_2} \prod_{j=1}^{2} (-1)^{m_j} \mathrm{e}^{-\frac{i\pi}{b^2} m_j} \prod_{j=1,4} \sin(2\pi m_1 b^{-1} p_j) \prod_{j=2,3} \sin(2\pi m_2 b^{-1} p_j)$$

$$\times \frac{b^2}{4\pi^2} \left( \frac{1}{(m_1 - m_2)^2} - \frac{1}{(m_1 + m_2)^2} \right). \tag{2.39}$$

In comparison, the exact worldsheet answer reads (2.35)

$$A^{(b)}_{\infty\text{-}\infty}(\boldsymbol{p}) = \sum_{m_1,m_2 \geqslant 1} \prod_{j=1}^{2} \frac{(-1)^{m_j}}{\sin(\pi m_j b^{-2})} \prod_{j=1,4} \sin(2\pi m_1 b^{-1} p_1) \prod_{j=2,3} \sin(2\pi m_2 b^{-1} p_j)$$
$$\times \frac{1}{\pi^2}\left(\frac{\delta_{m_1,m_2}}{(m_1-m_2)^2} - \frac{1}{(m_1+m_2)^2}\right). \tag{2.40}$$

Evidently, (2.39) and (2.40) are closely related. The two formulas have a different normalization factor, whose disagreement may be removed by shifting $S_0$ in the same way as for the three-point function. The other disagreements again have to do with the different treatment of the saddles labelled by $m$ and $-m$. After the replacement $\mathrm{e}^{-\frac{\pi i}{b^2}m_j} \to \sin(\frac{\pi i}{b^2}m_j)^{-1}$, we can relabel the sum by combining terms of $-m_j$ and $m_j$, which precisely leads to (2.40). Thus we seem to have explained all aspects of (2.35) from a gravitational path integral perspective, except for the combining of the saddles with label $m$ and $-m$. We will discuss this phenomenon a bit further in the discussion section 5.

# 3 The sphere and the Euclidean black hole

The analysis of the gravitational path integral is different for low genus and/or punctures which admit Killing vectors. The existence of non-trivial Killing vectors invalidates the argument in section 2.3. They contain a lot of interesting physics and we hence treat them separately here.

## 3.1 Solutions with Killing vectors

**Torus.** The torus with the flat metric can be endowed with a non-trivial Killing vector. However, this is inconsistent with (2.21b) since the second derivatives still vanish for this Killing vector, while $\Phi$ depends linearly on the coordinate. Thus we still need $\Phi = \frac{m}{2}$ with $m \in \mathbb{Z}$, but since the torus does not admit a metric with constant negative curvature, there is no solution on the torus.

**Two-punctured sphere.** For the sphere with two punctures, we can have a U(1) group of Killing vectors. This solution is known in the literature [43] and can be written as

$$\mathrm{d}s^2 = \frac{1}{f(r)}\,\mathrm{d}r^2 + f(r)\,\mathrm{d}\tau^2, \qquad \Phi_*(r) = r, \tag{3.1}$$

where $\tau$ is Euclidean time and

$$f(r) = \int_{r_0}^{r} \mathrm{d}r'\, W(r') = -\frac{\cos(2\pi r) - \cos(2\pi r_0)}{2\pi^2}, \tag{3.2}$$

where $W(r) \equiv \frac{1}{\pi}\sin(2\pi r)$ is the sine potential in (2.6). We have several choices for the range of $r$. Without loss of generality, we can take $r_0 \in [0, \frac{1}{2}]$. For $m \in \mathbb{Z}$, we then take

$$r \in \begin{cases} [\frac{m}{2} - r_0, \frac{m}{2} + r_0], & m \in 2\mathbb{Z}, \\ [\frac{m-1}{2} + r_0, \frac{m+1}{2} - r_0], & m \in 2\mathbb{Z}+1. \end{cases} \tag{3.3}$$

Here (in contrast to the more general surfaces of the previous discussion) we obtain a solution in $(-,-)$ signature for even $m$ and a solution in $(+,+)$ signature for odd $m$. The Ricci

scalar of the above metric is $\mathcal{R} = -2\cos(2\pi r)$ and can thus in particular change sign within a solution. Notice that even though the metric is unchanged when $m \to m + 2$, the dilaton shifts and thus constitutes a different solution as for the other surfaces above. However, since the Euler characteristic of the two-punctured sphere vanishes, the saddle point approximation around these solutions should be equivalent which explains why the two point function does not exhibit a sum over $m$. These solutions describe a Euclidean black hole solution with $\tau$ being the Euclidean time, which we identify periodically, $\tau \sim \tau + \beta$. We can relate $\beta$ and $r_0$ to the defect angle. We define a local coordinate $y = \sqrt{r - r_{\min}}$ or $y = \sqrt{r_{\max} - r}$ near the boundaries of the range of $r \in [r_{\min}, r_{\max}]$. In all cases, we obtain

$$
\mathrm{d}s^2 = (-1)^m \frac{4\pi}{\sin(2\pi r_0)} \left( \mathrm{d}y^2 + \frac{\sin(2\pi r_0)^2}{4\pi^2} y^2 \mathrm{d}\tau^2 \right). \tag{3.4}
$$

Due to the periodicity of $\tau$, this describes a conical defect with deficit angle

$$
2\pi - \alpha = \frac{\beta \sin(2\pi r_0)}{2\pi}. \tag{3.5}
$$

The two defect angles on the sphere are necessarily identical which leads to the appearance of the delta function in $\mathsf{A}_{0,2}^{(b)}(p_1, p_2)$.

**The sphere partition function.** We can further specialize this solution to obtain the solution on the once-punctured sphere where $\alpha = 0$ and thus $\beta$ is determined in terms of $r_0$. The path integral $\mathsf{A}_{0,1}^{(b)}(p_1)$ will lead to a delta function setting $p_1$ to zero defect angle. Thus, putting $\alpha = 0$ gives actually a smooth solution on the two-sphere, parametrized by the single modulus $r_0$ and $m$. The on-shell action is[2]

$$
S_{\text{o.s.}}[\Phi_*, g_*] = \frac{2\pi i m}{b^2}, \tag{3.6}
$$

and is of course independent of the modulus $r_0$. Notice in particular that the on-shell action agrees with the corresponding limit of (2.26). Indeed at special points of the moduli space parameterized by $r_0$ ($r_0 = 0$ or $r_0 = \frac{1}{2}$ depending on the parity of $m$), the solution (3.1) reduces to a solution with constant dilaton $\Phi_* = \frac{m}{2}$. The sphere partition function however turns out to diverge as is further discussed in Section 3.2. The reason is essentially the presence of moduli in the solution, even after fixing the $\mathrm{PSL}(2, \mathbb{C})$ gauge freedom.

**Euclidean black hole.** We can also further restrict the range in (3.3), which introduces boundaries, thus leading to the solution for the disk, the once-punctured disk (and the cylinder). The disk is of course nothing else than the Euclidean black hole and is the topology that is most widely discussed in the literature [44]. In particular, one can develop the usual black hole thermodynamics for these black hole solutions, as was done in general dilaton gravity theories in [45], see also [22] for sine dilaton theory.

Beyond the temperature $T = \frac{\sin(2\pi r_0)}{4\pi^2}$ that we already determined in (3.5), we can also compute the specific heat [2, 45, 46]. For the potential $W(\Phi) \equiv \frac{1}{\pi}\sin(2\pi\Phi)$, the specific heat is determined through the potential at the black hole horizon $r_{\min}$,

$$
C = -\frac{2\pi i}{b^2} \frac{W(\Phi(r_{\min}))}{\partial_\Phi W(\Phi(r_{\min}))} = -\frac{i}{b^2} \tan(2\pi r_{\min}) = \frac{i}{b^2}(-1)^m \tan(2\pi r_0). \tag{3.7}
$$

---

[2]In principle there is the possibility of an extra $(-1)^m$ on the right-hand side associated with the ambiguity of choosing a branch for $\sqrt{\det g_{\mu\nu}}$ from (3.1), given that the odd $m$ solutions are in $(-,-)$ signature. In accordance with (2.26) we have chosen the branch such that $\sqrt{\det(e^{2\rho_*}g_{\mu\nu})} = e^{2\rho_*}\sqrt{\det g_{\mu\nu}}$.

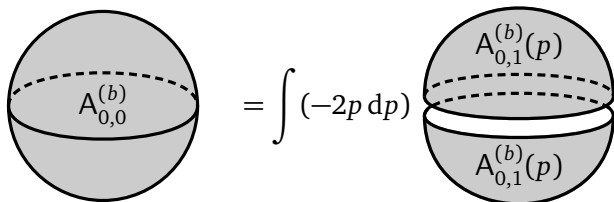

Figure 4: The sphere one-point amplitude $A_{0,1}^{(b)}(p)$ computes the 2d gravity path integral on the hemisphere, which prepares the Hartle-Hawking state. The norm of the Hartle-Hawking state is computed by the zero-point amplitude $A_{0,0}^{(b)}$, which is computed by gluing two hemispheres together.

In particular, the sign of the specific heat changes at $r_0 = \frac{1}{4}$. Recalling that $\frac{i}{b^2} > 0$ and assuming momentarily $m \in 2\mathbb{Z} + 1$ such that the metric is in $(+,+)$ signature, we see that large black holes with $0 \leqslant r_0 < \frac{1}{4}$ have negative specific heat and are thus thermodynamically unstable (meaning that the temperature increases as they evaporate, leading to a runaway evaporation), while small black holes with $r_0 > \frac{1}{4}$ can be thermodynamically stable. Thus the large black holes will evaporate until reaching $r_0 = \frac{1}{4}$ where the sign of the specific heat flips. We are not sure what the imprint of these two classes of black holes are on the observables we compute in the theory.

## 3.2 Two-sphere partition function

In our previous papers [25–27], we discussed the perturbative string amplitudes $A_{g,n}^{(b)}$, which as explained above can be viewed as a rigorous definition of the gravitational path integral of the 2d gravity theory (2.20). We will now discuss the exceptional cases $A_{0,0}^{(b)}$ and $A_{0,1}^{(b)}$ from a gravitational point of view.

**The one-point function as the HH-wavefunction.** In the context of de Sitter quantum gravity, the Hartle-Hawking state $|\Psi_{\mathrm{HH}}\rangle$ is prepared by the Euclidean gravitational path integral on the hemisphere [47]. In other words, in the momentum basis the Hartle-Hawking wavefunction is identified with the sphere one-point string amplitude $A_{0,1}^{(b)}$,

$$\Psi_{\mathrm{HH}}(p) = A_{0,1}^{(b)}(p). \tag{3.8}$$

The norm of the Hartle-Hawking wavefunction should be treated with caution. One may suspect that it is geometrically computed by the sphere path integral, which is obtained by stacking two hemispheres on top of each other as in figure 4. However, this is subtle because the conformal Killing vectors are potentially not treated correctly. We will see that both the norm of the Hartle-Hawking wavefunction as well as the sphere partition function diverge in this case and thus the equality formally holds.[3] Thus we should have

$$\||\Psi_{\mathrm{HH}}\rangle\|^2 = \int_0^{e^{-\frac{\pi i}{4}}\infty} (-2p\,\mathrm{d}p)\,|\Psi_{\mathrm{HH}}(p)|^2 \overset{?}{=} A_{0,0}^{(b)}. \tag{3.9}$$

It has been observed that the Hartle-Hawking state in some two-dimensional dilaton gravity theories diverges [3,49] and we will see that this is also the case in the sine dilaton theory (2.6).

---

[3]In more general situations the sphere partition function gets a phase as a consequence of the conformal mode problem of the Euclidean Einstein Hilbert action [48], complicating the relationship with the norm of the Hartle-Hawking state.

On the other hand models of de Sitter quantum gravity coupled to matter fields with a finite Hartle-Hawking wavefunction and a finite sphere partition function have been constructed [7, 8, 50]. The correct interpretation of the divergence of the sphere partition function for this dilaton theory remains an open question and we will discuss it further in the discussion section 5.

**Dilaton equation.** Computing $\mathsf{A}^{(b)}_{0,1}$ and $\mathsf{A}^{(b)}_{0,0}$ is somewhat subtle. In [25] we obtained the sphere partition function using the dilaton equations. In particular, we found

$$\mathsf{A}^{(b)}_{0,0} = \frac{1}{2b}\big(\mathsf{A}^{(b)}_{0,1}(\tfrac{1}{2}Q) + \mathsf{A}^{(b)}_{0,1}(\tfrac{1}{2}\hat{Q})\big) = \infty\,, \tag{3.10}$$

where

$$\mathsf{A}^{(b)}_{0,1}(p) = \frac{1}{2(1+b^2)}\big(\delta(p+\tfrac{1}{2}Q) - \delta(p-\tfrac{1}{2}Q)\big) + \frac{1}{2(1-b^2)}\big(\delta(p+\tfrac{1}{2}\hat{Q}) - \delta(p-\tfrac{1}{2}\hat{Q})\big), \tag{3.11}$$

and $\hat{Q} \equiv b^{-1} - b$.

**Sphere partition function from the path integral.** We will now evaluate the sphere partition function using the gravitational path integral. For this it is more convenient to go back to the formulation of the sine dilaton gravity in terms of two complex Liouville theories (2.2) and (2.4). We fix the background metric to be a round sphere with area $4\pi v$

$$d\tilde{s}^2 = 4v\frac{\mathrm{d}z\mathrm{d}\bar{z}}{(1+z\bar{z})^2}\,. \tag{3.12}$$

We furthermore rescale $\varphi \to \varphi/b$ leading to

$$S[\varphi] = \frac{1}{4\pi b^2}\int_{\mathrm{S}^2}\mathrm{d}^2 x\sqrt{\tilde{g}}\left(\tilde{g}^{\mu\nu}\partial_\mu\varphi\partial_\nu\varphi + \frac{2}{v}\varphi + \mathrm{e}^{2\varphi}\right), \tag{3.13}$$

where we also made the choice $\mu = \frac{1}{4\pi b^2}$ (see below (2.7)). We treat $S^*[\varphi]$ analogously. The equations of motion admit the saddle $\varphi_* = -\frac{1}{2}\log(-v)$, leading to the on-shell action and quadratic fluctuations

$$S_{\mathrm{o.s.}}[\varphi_*] = -\frac{1}{b^2}\left(1 + \log(-v)\right), \qquad S^{(2)}[\delta\varphi] = \frac{1}{4\pi b^2}\int_{\mathrm{S}^2}\mathrm{d}^2 x\sqrt{\tilde{g}}\,\delta\varphi(-\widetilde{\nabla}^2 - 2)\delta\varphi\,. \tag{3.14}$$

The sphere partition function of the complex Liouville string to quadratic order is thus given by

$$\mathcal{Z}^{\mathrm{S}^2}_{\mathrm{grav}} = \frac{v^{-\frac{13}{3}}}{\mathrm{vol}_{\mathrm{PSL}(2,\mathbb{C})}}\left|\int[\mathcal{D}\delta\varphi]\mathrm{e}^{-\frac{1}{4\pi b^2}\int_{\mathrm{S}^2}\mathrm{d}^2 x\sqrt{g}\,\delta\varphi(-\widetilde{\nabla}^2 - 2)\delta\varphi}\right|^2. \tag{3.15}$$

The exponent $-\frac{13}{3}$ is the Weyl anomaly contribution of the $\mathfrak{bc}$-ghost system. Going to Weyl gauge does not fully fix the gauge, but leaves the Moebius transformations $\mathrm{PSL}(2,\mathbb{C})$ as a residual gauge group that we need to divide by. Moebius transformations $f(z) \in \mathrm{PSL}(2,\mathbb{C})$ act non-trivially on the Weyl factor $\varphi$

$$\varphi(z,\bar{z}) \to \varphi(f(z), \overline{f(z)}) + \frac{Q}{2}\log|f'(z)|^2\,, \tag{3.16}$$

and leave the Liouville action (3.13) invariant [7], thus leading to an additional infinite $\mathrm{vol}_{\mathrm{PSL}(2,\mathbb{C})}$ upstairs. This already indicates that we need to treat (3.15) with care. We can further expand the fluctuation $\delta\varphi$ into eigenfunctions of the two-sphere Laplacian $\widetilde{\nabla}^2$

$$\delta\varphi = \sum_{l,m}\delta\varphi_{l,m}Y_{l,m}(\Omega), \quad \text{where } -\widetilde{\nabla}^2 Y_{l,m}(\Omega) = l(l+1)Y_{l,m}(\Omega). \tag{3.17}$$

$\Omega$ is a point on the two-sphere and $Y_{l,m}(\Omega)$ with $l \geqslant 0$ and $-l \leqslant m \leqslant l$ are the spherical harmonics, which we choose to be real valued; $\delta\varphi_{l,m} \in \mathbb{C}$.[4] From this it is clear that the three-fold degenerate $l = 1$ modes are zero modes of (3.15) corresponding respectively to three of the six conformal Killing vectors on $S^2$ that are not isometries, i.e. part of SO(3), of the round two-sphere. We can fix the infinite volume of PSL(2,$\mathbb{C}$) using a Faddeev-Popov gauge fixing, following [7,51]. As a gauge fixing condition we then choose $\delta\varphi_{1,m} = 0$ for $m = -1,0,1$. We thus obtain

$$\int \frac{[\mathcal{D}\delta\varphi][\mathcal{D}\delta\varphi^*]}{\mathrm{vol}_{\mathrm{PSL}(2,\mathbb{C})}} e^{-2\,\mathrm{Re}\,S^{(2)}[\delta\varphi]} = \int \frac{[\mathcal{D}\delta\varphi][\mathcal{D}\delta\varphi^*]}{\mathrm{vol}_{\mathrm{SO}(3)}} \Delta_{\mathrm{FP}} \prod_{m=-1,0,1} \delta(\delta\varphi_{1,m}) e^{-2\,\mathrm{Re}\,S^{(2)}[\delta\varphi]}, \quad (3.19)$$

where $\Delta_{\mathrm{FP}}$ denotes the Faddeev-Popov determinant. In particular $\delta\varphi$ is complex valued and hence the combined theory (3.15) has six zero modes. While three of these can be fixed using the three non-compact directions of PSL(2,$\mathbb{C}$) the other three remain unfixed and lead to a divergent two-sphere partition function. Since the two-sphere is central to the Gibbons-Hawking de Sitter entropy [52,53] proposal it would be interesting to better understand what consequences we should learn from this. A divergent sphere partition function in JT dS has been anticipated in [3], and calculated using the relation to the $(2,p)$−matrix model in [39] as well as by counting zero modes in [49]. The sphere partition function also diverges in the Virasoro minimal string [15], whereas it seems to be finite in other two-dimensional models [7,8,50,54]. Our finding is in tension with [55], where it was claimed that the divergence can be removed with the help of the Coulomb gas formalism.

# 4 (A)dS$_2$ quantum gravity

We will now discuss some of the lessons that we learned by comparing the gravitational path integral with the exact worldsheet answer.

## 4.1 Genus expansion

The genus expansion of the gravitational partition functions $Z_n^{(b)}(S_0; \mathbf{p})$ is alternating in sign. Indeed, it was discussed in [27] that the effective string coupling that governs the asymptotics of the genus expansion takes the form

$$g_s^{\mathrm{eff}} = \frac{8\sin(\pi b^2)\sin(\pi b^{-2})}{b^{-2} - b^2} \in i\mathbb{R}. \quad (4.1)$$

It in particular takes purely imaginary values. The oscillating nature of the genus expansion of a related model, dS JT gravity, was also motivated in [12]. However, declaring that the perturbative expansion becomes oscillating for the analytic continuation of AdS gravity to dS gravity typically spells doom on the non-perturbative completion of the theory.

To make this point, consider the symmetric orbifold $\mathrm{Sym}^N(\mathbb{T}^4)$, which describes the CFT dual of AdS$_3 \times S^3 \times \mathbb{T}^4$ at the tensionless point in moduli space [56,57]. From the Brown-Henneaux central charge formula [58], $c = \frac{3\ell}{2G_N^{(3)}}$ one sees that analytic continuation to dS$_3$ formally gives an imaginary central charge. This would lead one to suggest that the symmetric orbifold with *imaginary N* is dual to a de Sitter background. This can easily be implemented

---

[4]The complex Liouville string combines two $c = 13 \pm i\lambda$ Liouville theories. We expand also the second Liouville field in a basis of spherical harmonics

$$\delta\varphi^* = \sum_{l,m} \delta\varphi_{l,m}^* Y_{l,m}(\Omega), \quad (3.18)$$

where we used $\delta\varphi = \delta\varphi^*$ and the reality condition $Y_{l,m}(\Omega)^* = Y_{l,m}(\Omega)$ of the spherical harmonics.

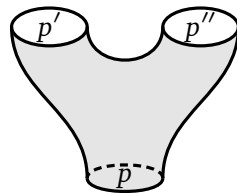

Figure 5: The sphere three-point amplitude $A_{0,3}^{(b)}(p, p', p'')$ may be interpreted as the leading contribution to the transition amplitude between one initial universe labelled by $p$ and two final universes labelled by $p'$ and $p''$.

in a perturbative $\frac{1}{N}$ expansion, where we can replace $N$ by $iN$ termwise and which renders the $\frac{1}{N}$ expansion oscillating. However, there is presumably no non-perturbatively defined CFT where $N$ is imaginary.

In the present case we do however expect that there is a sensible non-perturbative completion of the theory since we started with a *bona fide* string construction. In particular, aspects of this non-perturbative completion were tested in [27] and exhibit substantial qualitative differences from the structure of non-perturbative corrections found in, say, ordinary JT gravity or the Virasoro minimal string. It is tempting to speculate that such a non-perturbative completion only exists thanks to the coexistence of both AdS and dS vacua in the model.

## 4.2 Universe transitions

**Transition amplitudes.**   We now discuss the question of what these computations imply for the observables in de Sitter quantum gravity. The partition functions $Z_{g,n}^{(b)}$ (which we take in the following to be given by the exact formula (2.35)) form a complete set of observables in the model. Cosmologically, they compute the time evolution of the cosmological wave function. The number of components in a spatial slice is not necessarily constant and new baby universes [59–61] can form or disappear. See figure 5 for a representation of the transition amplitude between one initial universe and two final universes computed by $A_{0,3}^{(b)}(p, p', p'')$. We can in particular consider the situation with no past boundary, which simply prepares cosmological states such as the Hartle-Hawking state discussed above. In general, after summing over topologies, this is consistent with the no-boundary proposal [47].

When computing norms of such states, we form closed manifolds with a spatial slice of the appropriate topology and then sum over all possible manifolds. Such a sum in general should also contain bra-ket wormholes connecting the part of the topology preparing the bra state with the part preparing the ket state [62].

This is so far analogous to the situation for AdS$_2$, see [63–66] for related discussions. In particular, the ensemble average in the dual matrix model appears because of spacetime wormholes via the Coleman-Giddings-Strominger mechanism [59–61].

**Vacua.**   The novelty compared with previous dualities between theories of two-dimensional gravity and matrix models is the interplay with the different vacua of the theory. In the semiclassical interpretation of the theory where we sum over different saddles, the initial part of the universe can be either in an AdS or a dS saddle. As discussed above, the gravitational path integral includes transitions between both types of universes. We should note that the specification of the initial vacuum is *not* part of the initial data since we a priori have to sum over all saddles. We can nevertheless go ahead and discuss individual contributions to the gravitational path integral even though this is not a well-defined observable in the full theory. Let us consider for concreteness the evolution of a single-universe state to a single-universe

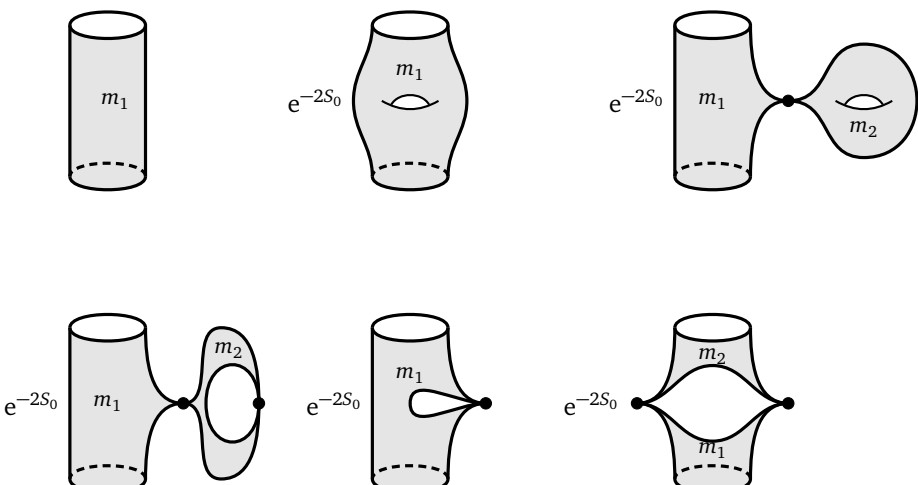

Figure 6: Contributions to the two-boundary gravitational path integral, classified by stable graphs corresponding to generations of the appropriate bordered Riemann surface at each order in perturbation theory, up to $\mathcal{O}(e^{-2S_0})$. Starting at $e^{-2S_0}$ there are contributions from infinitely many $AdS_2$ and $dS_2$ saddles (corresponding to odd and even values of the integers $m_i$, respectively). The configurations in the bottom right include in particular contributions from saddles where one boundary is in an AdS universe and the other is in a dS universe. Collectively the sum over colors of all but the top-left configuration make up the torus two-point string amplitude $A_{1,2}^{(b)}$.

state. This has a leading contribution at $\mathcal{O}(1)$ in $S_0$ given by $A_{0,2}^{(b)}$, which does not allow transitions between different universes. However, starting from $A_{1,2}^{(b)}$, such transitions do occur. They are hence suppressed as $e^{-2S_0}$. One can view this as an indication that the vacua are all metastable. It seems however difficult to directly tie this to the discussion of stability of de Sitter saddles in higher dimensions because of the different nature of the dilaton-gravity action [67].

**Third-quantized picture.** We finally mention that the most complete way to formulate the gravity theory is in a third-quantized picture. Such a third-quantized picture is essentially the string field theory treatment of the worldsheet theory. In the present case, there is a very explicit way to formulate the string field theory. It is given as a 2d Kodaira-Spencer theory on the spectral curve, obtained by compactifying 6d Kodaira-Spencer theory of the topological B-model to 2d [68]. The basic field in 2d becomes a chiral boson with $\mathbb{Z}_2$-twists at the branch points of the spectral curve. Its Hilbert space precisely corresponds to the Fock space of the baby universes. This perspective was worked out for JT gravity in detail in [66,69]. We should however note that the formulation of this theory presumes the knowledge of the duality of the gravity theory with the matrix integral that we presented in [26].

## 5 Discussion

We will now discuss a few open questions and future directions.

**Relation to DSSYK.** Our discussion of the complex Liouville string is superficially similar to recent discussions in [22,23] and [10,11], where the bulk theory was presented either as two coupled Liouville theories or as a sine dilaton gravity.

However, while these discussions involve the same bulk theory, the conclusions are rather different and orthogonal to our considerations. The authors propose a relation of this theory on the disk to double-scaled SYK (DSSYK), which can be solved via so-called chord diagrams [70]. We consider the correspondence of sine dilaton gravity to the matrix model by relating it to the complex Liouville string that we proposed in this series of papers to be on much firmer ground as we have essentially derived it. The conjectured relation to DSSYK does not seem to be reflected in our exploration, but this is partly due to the fact that we have emphasized a different set of observables. There are a few obvious problems and clashes with the lore of this theory:

1. DSSYK is only defined in the strict large $N$ scaling limit and does not have a genus expansion. Thus only the leading order terms can be matched. Hence the main observables are disk correlators with boundary insertions. In particular, we don't expect DSSYK to have any knowledge about the perturbative string amplitudes $A_{g,n}^{(b)}(p_1,\ldots,p_n)$.

2. The boundary condition considered in [10] with two FZZT boundary conditions on both Liouville theories whose FZZT parameters are complex conjugates does not seem to be special from our point of view. See [27] for further discussion of conformal boundary conditions in the complex Liouville string.

3. The density of states of DSSYK is expressed in terms of a Jacobi theta function. It is in particular distinct from the density of states that appears in the matrix integral of the complex Liouville string, see [26, eq. (3.9)]. The topological recursion based on the density of states of DSSYK produces discrete point-counting analogues of the Weil-Petersson volumes [71], which again bears little resemblance to the perturbative amplitudes $A_{g,n}^{(b)}(\mathbf{p})$.

4. Disk correlation function of DSSYK are controlled by the representation theory of the quantum group $\mathcal{U}_q(\mathfrak{su}(1,1))$.[5] We have not identified any signatures of this quantum group in the complex Liouville string. Moreover, it is expected that the bulk dual of DSSYK would reflect the discrete and non-commutative geometry of the chord diagrams [73–75]; but, the present bulk theory again seems to be perfectly smooth.

It has also been suggested recently in [28] that some additional gauging of a discrete symmetry has to be performed on the gravity side to reproduce a gravity theory dual to DSSYK. In the rigorous formulation of sine dilaton gravity in terms of two copies of Liouville theory that we have advocated for in this series of papers as well as in the path integral formulation, such a discrete symmetry *does not exist*. Thus that proposal seems fundamentally incompatible with our findings.

**Saddle recombination.** Let us emphasize one important technical finding. We learned by comparing the gravitational path integral with the exact answer extracted from the worldsheet that the saddle point structure of the gravitational has the unexpected feature of combining the saddle with dilaton value $\Phi = \frac{m}{2}$ with the saddle with dilaton value $\Phi = -\frac{m}{2}$. This seems almost like we gauged $\Phi \sim -\Phi$, but such a gauging is of course not possible because the action (2.8) is antisymmetric under $\Phi \to -\Phi$ rather than symmetric. Therefore it would be interesting to understand the precise mechanism from the path integral that leads to this phenomenon and possible generalizations to gravitational path integrals in other situations.

---

[5]Instead, we expect the relevant quantum group to be a complex quantum group denoted by $\mathcal{U}_q(\mathfrak{sl}(2,\mathbb{C})_{\mathbb{R}})_S$ in [72].

**Two-sphere partition function.**   One issue arising in dilaton-gravity theories is the divergence of some low $(g, n)$ string amplitudes such as the sphere partition function and the sphere one-point function. In models with a positive cosmological constant the sphere one-point function is the Hartle-Hawking wavefunction [47], whereas the sphere partition function captures the de Sitter entropy [52,53]. Moreover in certain instances it can be interpreted as the norm of the Hartle-Hawking wavefunction [3]. This is in contrast to models with finite Hartle-Hawking wavefunction and sphere partition function [7–9]. There are several interpretations of this result. One may view this as an indication that the no-boundary state is in fact not a natural wavefunction of the universe. We argue in [76] for such a resolution in 3d gravity, where the no-boundary wavefunction has a similar problem for sufficiently complicated Cauchy slices. Another possible interpretation is that the path integral computing the sphere partition function should in some way be modified so as to lead to a finite de Sitter entropy, but we have no concrete proposal how to achieve this on the bulk side. A prescription on the matrix side that we have also employed in [76] in the context of $dS_3$ is to cut off the eigenvalue density in the matrix model at the first zero, which leads to effectively a finite number of eigenvalues. Lastly, one might think that the divergent two-sphere partition function truly means that the entropy is infinite, perhaps because of the coexistence of an infinite number of de Sitter and anti-de Sitter vacua.

**Stability of the vacua and alternating saddles.**   In this work we have seen that the complex Liouville string admits a semiclassical description in terms of two-dimensional sine dilaton gravity. Intriguingly, when considered on a surface with negative Euler characteristic, this model admits an infinite series of classical solutions which alternate between positive and negative curvature. The string amplitudes of the complex Liouville string include contributions from all of these saddles; at higher orders in $e^{-S_0}$, they include contributions from nodal surfaces which involve transitions from $dS_2$ universes to $AdS_2$ universes (and vice-versa), and between different universes of the same curvature. One may take this as a suggestion that all the vacua are metastable, but it is difficult to address this question directly since the initial vacuum is not part of the initial data specified in computing the string amplitudes; the latter involve a sum over all saddles. It is tempting to speculate that the additional $AdS_2$ saddles are ultimately needed for an ultraviolet-complete description of $dS_2$ quantum gravity.

**2d black hole and relation to 4d.**   In this work we studied the sine-dilaton gravity theory on Riemann surfaces excluding the disk. On the disk the dilaton profile is no longer constant but the equations of motion (2.21b) and (2.21a) admit the solution (3.1). The Ricci scalar associated to this metric exhibits regions of a de Sitter, anti-de Sitter and Minkowski spacetime. It would be interesting to understand whether this model could be obtained from a dimensional reduction of a four-dimensional black hole much like $AdS_2$ and $dS_2$ JT gravity. In particular the near-extremal, near-horizon limit geometry of the RN de Sitter black hole is either $AdS_2 \times S^2$, $dS_2 \times S^2$ (the Nariai limit) or $Mink_2 \times S^2$.

# Acknowledgments

We would like to thank Dionysios Anninos, Alessandro Fumagalli, Adam Levine, Juan Maldacena, Jan Pieter van der Schaar, Boris Post, Erik Verlinde, Herman Verlinde, Edward Witten. We especially thank Victor Rodriguez for initial collaboration and discussions about related topics.

**Funding information** We thank l'Institut Pascal at Université Paris-Saclay, with the support of the program "Investissements d'avenir" ANR-11-IDEX-0003-01, and SC thanks the Kavli Institute for Theoretical Physics (KITP), which is supported in part by grant NSF PHY-2309135, for hospitality during the course of this work. SC is supported by the U.S. Department of Energy, Office of Science, Office of High Energy Physics of U.S. Department of Energy under grant Contract Number DE-SC0012567 (High Energy Theory research), DOE Early Career Award DE-SC0021886 and the Packard Foundation Award in Quantum Black Holes and Quantum Computation. LE is supported by the European Research Council (ERC) under the European Union's Horizon 2020 research and innovation programme (grant agreement No 101115511). BM gratefully acknowledges funding provided by the Sivian Fund at the Institute for Advanced Study and the National Science Foundation with grant number PHY-2207584.

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
