# Peer review of "The complex Liouville string: the gravitational path integral"

_SciPost Physics, doi:SciPost Phys. 19, 115 (2025)_

## Round 1 · Referee Report · Anonymous (Referee 1) · 2025-5-22

Strengths

  1. Quantitative analysis of the 2d gravitational path integral that admits de Sitter backgrounds, guided by string theory and duality to a matrix model.
  2. Clear and pedagogical exposition.

Weaknesses

  1. Most of the technical results in the paper rely on the results obtained already in previous papers.

Report

The paper discusses the 2d dilaton gravity with a sine potential, a model discussed also by other groups as a solvable model of gravity that admits de Sitter vacuum. Based on the complex Liouville string they have developed in the preceding papers, they discussed the relation to the sine-dilaton gravity path integral clarifying several aspects of the gravitational path integral and shedding new lights on it. The results are certainly interesting and illuminating although most of the technical computations performed in the paper rely on the results of their previous papers and it raises a question of whether it could have been incorporated into one of their previous papers. That said, I think the paper is worth to be published.

Requested changes

  1. I found the way (2.6) is presented is slightly misleading; initially the authors writes it as if it can be directly derived from the complex Liouville action and later says at the end of section 2.1 that you cannot actually derive the second line of (2.6). It would be better to improve the presentation.
  2. Worldsheets with nodal points show up often in the double-trace deformation of matrix models. Is there any relation?

Recommendation

Ask for minor revision

  • validity: high
  • significance: good
  • originality: good
  • clarity: high
  • formatting: perfect
  • grammar: perfect

Author:  Beatrix Mühlmann  on 2025-10-02  [id 5885]

(in reply to Report 1 on 2025-05-22)

We thank the referee for their careful reading. We have adjusted equation (2.6) and the surrounding phrasing. To the best of our knowledge, there is no immediate relation to double-trace deformations in our model.

---

## Round 1 · Referee Report · Anonymous (Referee 2) · 2025-8-14

Strengths

  1. Novel type of worldsheet theory that is solvable
  2. Interesting link to JT gravity with sine potential
  3. Potential applications to AdS_2 and dS_2 quantum gravity
  4. Clearly presented analysis

Weaknesses

  1. Bulk unitarity properties of deformed JT theories unclear

Report

The paper discusses and puts on more firm footing a solvable worldsheet theory, given by combining complexified Liouville theories with complex conjugate central charges. The theory finds a surprising number of potential applications to AdS_2, dS_2, and even dS_3 gravity, displaying its richness. It is clearly presented, and I support its publication.

Recommendation

Publish (easily meets expectations and criteria for this Journal; among top 50%)

---

## Round 1 · Referee Report · Anonymous (Referee 3) · 2025-8-15

Report

In this work the authors continue their explorations of the complex Liouville string (CLS). To remind, the worldsheet theory is two copies of Liouville theory, one at complex central charge $c = 13 + i \nu$ and the other at $c^*$, coupled to the usual worldsheet conformal gravity. In other works the authors (together with Rodriguez) have given convincing evidence that this model has a two-matrix model dual and moreover computes certain amplitudes in 3d de Sitter gravity.

Here the main goal is to understand the physics of the worldsheet dilaton gravity and the rules of the game for its path integral. They do so through worldsheet gravity computations of string amplitudes that are then matched to exact answers. This leads to surprising non-perturbative rules that would have been difficult to anticipate from a perturbative worldsheet approach.

Two interesting results are that (1) the effective string coupling is pure imaginary, and (2) that the model has perturbative AdS and dS (well, Euclidean AdS in (-,-) signature) vacua as well as transitions between them.

I think the paper is absolutely lovely and should be published quickly (and apologize for how I have delayed that process by being slow to give this report). My only comments are:

  1. The sphere partition function of de Sitter JT gravity, which was discussed a few times in the manuscript, was not really computed in the 2019 papers of Maldacena/Turiaci/Yang and Cotler/Jensen/Maloney. Both of those consider the unnormalized probability distribution of the HH state, which diverges, suggesting that the norm (assumed to be the sphere amplitude) diverges. In AdS JT the sphere was not computed until Mahajan/Stanford/Yan; it was then computed in dS JT (somewhat trivially) using the dS measure in the Cotler/Jensen 2024 paper.

  2. It goes beyond the goals of the manuscript, but I think it should be noted somewhere in the manuscript that it is an open problem to understand how to obtain worldsheets with asymptotically de Sitter regions, not merely Euclidean spaces in (-,-) signature with finite-size boundaries, assuming that this can even be done.

Recommendation

Publish (surpasses expectations and criteria for this Journal; among top 10%)

  • validity: top
  • significance: high
  • originality: high
  • clarity: high
  • formatting: perfect
  • grammar: excellent

Author:  Beatrix Mühlmann  on 2025-10-02  [id 5883]

(in reply to Report 3 on 2025-08-15)
Category:
remark
correction

We thank the referee for their careful reading and for clarifying the correct historical references regarding the sphere partition function. The references have been adjusted accordingly below eq. 3.19.

---

## Editorial Decision

published